# 1.34 µm Q-Switched Nd:YVO_4_ Laser with a Reflective WS_2_ Saturable Absorber

**DOI:** 10.3390/nano9091200

**Published:** 2019-08-26

**Authors:** Taijin Wang, Yonggang Wang, Jiang Wang, Jing Bai, Guangying Li, Rui Lou, Guanghua Cheng

**Affiliations:** 1School of Physics and Information Technology, Shaanxi Normal University, Xi’an 710119, China; 2Department of Physics, Taiyuan Normal University, Taiyuan 030031, China; 3State Key Laboratory of Transient Optics and Photonics, Xi’an Institute of Optics and Precision Mechanics, Chinese Academy of Sciences, Xi’an 710119, China; 4Electronic Information College, Northwestern Polytechnical University, Xi’an 710072, China

**Keywords:** WS_2_, saturable absorbers, Langmuir–Blodgett technique, Q-switched laser

## Abstract

In this work, a Tungsten disulfide (WS_2_) reflective saturable absorber (SA) fabricated using the Langmuir–Blodgett technique was used in a solid state Nd:YVO_4_ laser operating at 1.34 µm. A Q-switched laser was constructed. The shortest pulse width was 409 ns with the repetition rate of 159 kHz, and the maximum output power was 338 mW. To the best of our knowledge, it is the first time that short laser pulses have been generated in a solid state laser at 1.34 µm using a reflective WS_2_ SA fabricated by the Langmuir–Blodgett method.

## 1. Introduction

Saturable absorbers (SA) have been used as a switching element to generate short pulses in passively Q-switched lasers. It is mainly represented by transition metal ions–doped bulk crystals like Cr^4+^:YAG and V^3+^:YAG [1,2,3,4,5], semiconductor materials like the Semiconductor Saturable Absorbing Mirror (SESAM) [6,7,8,9], and two-dimensional (2D) materials [10,11,12,13,14,15,16,17].

The fabrication method of switching elements is very important and determines the performances of the Q-switching lasers. The Langmuir–Blodgett (LB) technique is a convenient and low-cost method for preparing ultrathin nano materials films [18].

Two-dimensional materials have been widely used in laser applications [19,20,21,22,23] due to their simple structure and remarkable wide spectral band [24,25,26]. 2D atomically thin Tungsten disulfide (WS_2_) nanosheets exfoliated from bulk counterparts have shown exotic electronic and optical properties, such as indirect-to-direct bandgap transition with a reducing number of layers (the indirect band gap is ~1.3 eV and the direct band gap of its monolayer form is up to 2.1 eV), high carrier mobility, and strong spin–orbit coupling due to their broken inversion symmetry, which have enabled wide potential applications in viable photonic and optoelectronic devices [27,28,29]. As a kind of 2D material, WS_2_ has been successfully developed to produce short pulses in lasers with various wavelengths such as 1.06 µm, 1.53 µm, 1.65 µm, and 3 µm [30,31,32,33,34].

In this paper, the LB technique was used to coat few-layer WS_2_ onto a silver-coated mirror. In this way, a low-cost reflective WS_2_ saturable absorber (SA) was prepared. Based on the reflective WS_2_ SA, a passive Q-switched solid state Nd:YVO_4_ laser was constructed, which generated short pulses at 1.34 µm. The maximum average Q-switched output power of 338 mW was obtained with the pulse repetition rate of 159 kHz, corresponding to the single pulse energy of 2.13 µJ and peak power of 5.20 W, respectively. The results indicate that the WS_2_ can be fabricated by the Langmuir–Blodgett method and used as a Q-switch element in solid state lasers to generate short pulses at 1.34 µm.

## 2. Materials and Methods

### 2.1. WS_2_ Saturable Absorber Fabrication

The few-layer WS_2_ suspension was fabricated from the bulk WS_2_ by liquid phase exfoliation. A bulk WS_2_ (from XF NANO Inc., Nanjing, China) was ultrasonicated for 24 h and centrifuged for 20 min to get the aqueous solution with the concentration of 2 mg/mL.

The methanol, chloroform, and as-prepared WS_2_ supernatant with the volume ratio of 1:1:4 was prepared and ultrasonicated for 15 min. The Raman spectrum of the WS_2_ silicon wafer was measured by a Raman spectrometer (LabRam confocal Microprobe system, Horiba Jobin Yvon, Paris, France).

The reflective WS_2_ saturable absorber, a silver mirror coated with WS_2_ saturable absorber by the Langmuir–Blodgett technique, and the Langmuir–Blodgett system (JML04C1, 2017JM7085, Powereach, Shanghai, China), are shown in Figure 1. The silver mirror was composed of a 180 nm silver film and a 20 nm silica protection film evaporated on a quartz plate by the electron beam aided evaporation technique.

The prepared WS_2_ solution was dipped into a trough containing deionized water and spread on the surface of the cell. The trough contained 200 mL deionized water and the pH of deionized water was 7.0. The instillation stopped until the pressure of the surface, measured by a force transducer, reached 35 mN/m steady. Then the silver mirror, pre-inserted into the deionized water, was pulled up slowly. At the same time, the surface of the liquid was compressed by two mobile barriers with the speed of 4.85 mm/min under the control of a motor. After the silver mirror coating, the WS_2_ films were pulled out from the liquid completely and then dried at 80 °C for 10 min. The reflective WS_2_ SA was fabricated successfully.

### 2.2. Characterization of WS_2_ Saturable Absorber

The surface of the WS_2_ films was characterized by a scanning electron microscope (SEM, Nova NanoSEM Training-X50 series, FEI, Eindhoven, The Netherlands) and the thickness of the WS_2_ films was characterized by an atomic force microscope (AFM, Dimension Icon, Bruker Nano Inc., Mannheim, Germany).

A spectrophotometer (Perkin-Elmer, UV-Lambda 1050, Downers Grove, IL, USA) was used to measure the linear optical reflectivity curve of the reflective WS_2_ saturable absorber and the nonlinear optical characteristics of the reflective WS_2_ SA were measured by a balanced twin-detector measurement technique, which was described in [35]. The pump source for the nonlinear optical measurement was a self-made acoustic-optically Q-switched Nd:YVO_4_ laser with the pulse of 40 ns and a repetition rate of 10 kHz at 1.34 µm.

### 2.3. Laser Cavity

Figure 2 shows the schematic setup of the Nd:YVO_4_ passively Q-switched laser with a reflective WS_2_ SA at 1.34 µm.

There was a 3 × 3 × 10 mm a-cut Nd:YVO_4_ crystal with a Nd^3+^ ions doping concentration of 0.5 at.%, and the 808 nm anti-reflective films were coated onto both sides of its ends. Water-cooled equipment was used to maintain the temperature of the laser crystal at 12 °C. The crystal was wrapped with indium foils contacted tightly with copper heat sink.

A fiber-coupled laser diode (LD) with the maximum output power of 50 W and the central wavelength of 808 nm was used as the pump source. The pump light was focused on the Nd:YVO_4_ crystal with a pump spot diameter of 400 µm after passing through a 1:1 coupling lens, a flat mirror, and a concave output coupler (OC). The flat mirror was coated with anti-reflective film at 808 nm and high-reflective film (R > 99.9%) at 1342 nm. The output coupler with the curvature radius of r = 100 mm had a transmission of 5%. The length of the cavity was about 14 mm. it was set up with a reflective WS_2_ saturable absorber and an output coupler. The distance from the laser crystal to the output coupler and the reflective WS_2_ saturable absorber were 1 and 3 mm, respectively.

The average output power of the Q-switched laser at 1.34 µm can be measured accurately by a power meter. The data of the output Q-switched pulse repetition rate and duration were recorded by a digital oscilloscope (Rohde & Schwarz, RTO1014, Munich, Germany) with a photodetector (Thorlabs, DET08C/M, Munich, Germany). A laser spectrum analyzer (YOKOGAWA, AQ6370D, Suzhou, China) was employed to record the spectrum.

## 3. Results and Discussion

### 3.1. Characteristics of WS_2_ Saturable Absorber

Figure 3 shows the Raman spectrum of the few-layer WS_2_ excited by a 532 nm laser source. The locations of two characteristic Raman active vibration modes, viz., $E_{2g}^{1}$ (in-plane) at 356.3 cm^−1^ and $A_{g}^{1}$ (out-of-plane) at 417.0 cm^−1^, should be in agreement with other reported few-layer WS_2_ [36].

The thickness and surface roughness of the WS_2_ SA films are shown in Figure 4. The image of the atomic force microscope (AFM) is shown in Figure 4a. The thickness of the WS_2_ films is about 5 nm, and the surface roughness is less than 1 nm in Figure 4b. In addition, the surface of the WS_2_ films is shown by the scanning electron microscope (SEM) image in Figure 4c. In short, it was determined that the surface of WS_2_ films was very uniformed. Moreover, we could estimate the size of the particle of the WS_2_ films, it was about dozens of micron.

The reflection spectrum of the reflective WS_2_ saturable absorber is measured by a wavelength range from 1000 to 1400 nm in Figure 5. It shows the reflectivity of the sample is about 64.8%.

The nonlinear absorption saturation characteristics of the WS_2_ saturable absorber is displayed in Figure 6. The schematic diagram of nonlinear optical absorption measurement is depicted as the inset.

The reflectivity of the WS_2_ saturable absorber versus different incident pulse energy intensities was recorded. The data of the reflectivity is depicted as dots in Figure 7, and fitted by the following equation [35]: T(I)=1−ΔTexp(−I/Isat)−Tns, where *T(I)* is the reflectivity of the reflective WS_2_ saturable absorber, ∆*T* is the modulation depth, *I_sat_* is the saturable intensity, and *T_ns_* is the non-saturable loss. The modulation depth and the saturation intensity of the reflective WS_2_ SA were simulated to be 24.5% and 71.9 kW/cm^2^, respectively.

### 3.2. WS_2_ Q-Switched Laser

Firstly, the operation of the continuous wave (CW) Nd:YVO_4_ laser with a output coupler and a high-reflective mirror was investigated. The relationship between continuous wave laser output power and pump power is observed in Figure 7a. As shown in Figure 7a, the pump power threshold of the continuous wave laser and the slope efficiency of the almost linear relationship are 37 mW and 22.8%, respectively. No self-Q-switched pulse was observed in the generation of the continuous wave laser.

The operation of the passively Q-switched (QW) laser was effected after replacing the HR mirror with the reflective WS_2_ saturable absorber. The data of the Q-switched average output power are shown in Figure 7a. The Q-switched operation remained unchanged when the pump power increased from 1.84 to 2.83 W, and the Q-switched laser output power changed from 132 to 338 mW correspondingly, with a slope efficiency of 19.9%.

The pulse widths and repetition rates were recorded synchronously. The evolution of the pulse width and repetition rates of the pump power is presented in Figure 7b. Based on the data of the output Q-switched laser, it was directed to get the single pulse energies and peak powers. As displayed in Figure 7c, the maximum single pulse energy of 2.13 μJ and pulse peak power of 5.20 W was obtained when the pump power was 2.83 W.

Three different individual pulses under different pump powers were depicted in Figure 8a,b to show the evolution of the pulse width and repetition rates with the pump power visually. It demonstrated that the pulse width decreased from 720 to 409 ns of and the repetition rate increased from 115 to 159 kHz with the increase of the pump power from 1.84 to 2.83 W. A slight jitter of the pulse trains is observed from Figure 8a. The jitter was primarily caused by the thermal instability of the reflective SA under long time laser illumination. It was also possible that the instability of the laser cavity attributed to the thermal lens effect from the laser crystal to give rise to the jitter. The shortest pulse duration of 409 ns was obtained and is displayed in Figure 8b. The QW spectrum was measured and is shown in Figure 8c. The central wavelength (λ_c_) was 1342 nm with the bandwidth of 0.12 nm.

## 4. Conclusions

In this work, we presented a new kind of reflective WS_2_ saturable absorber fabricated using the Langmuir–Blodgett technique and constructed, for the first time, a passively Q-switched Nd:YVO_4_ solid state laser at 1.3 μm with the absorber. It had ideal characteristics in thickness and the uniformity of the nanomaterials. The shortest duration was achieved with pulse width of 409 ns, and the highest peak power was 5.20 W. These results indicate that a reflective WS_2_ saturable absorber with perfect characteristics fabricated using the Langmuir–Blodgett technique can be a promising optical modulator to generate short pulses at 1.3 μm.

## Figures and Tables

**Figure 1 nanomaterials-09-01200-f001:**
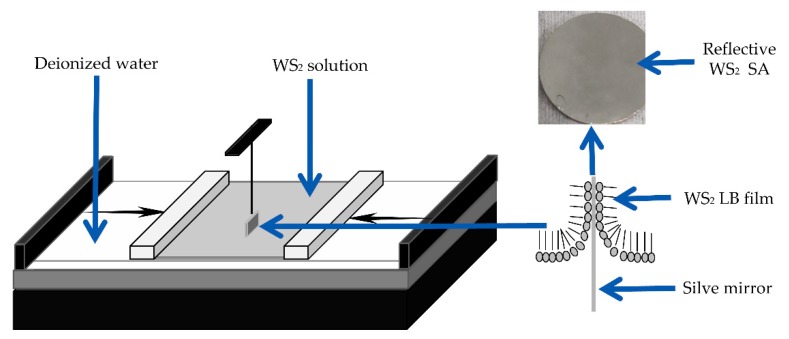
The fabrication of the reflective WS_2_ saturable absorber (SA) by the Langmuir–Blodgett (LB) system. Inset: the reflective WS_2_ saturable absorber.

**Figure 2 nanomaterials-09-01200-f002:**
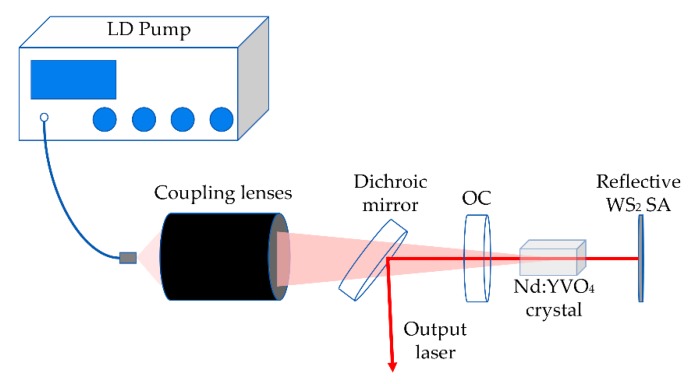
The schematic setup of the Nd:YVO_4_ passively Q-switched laser.

**Figure 3 nanomaterials-09-01200-f003:**
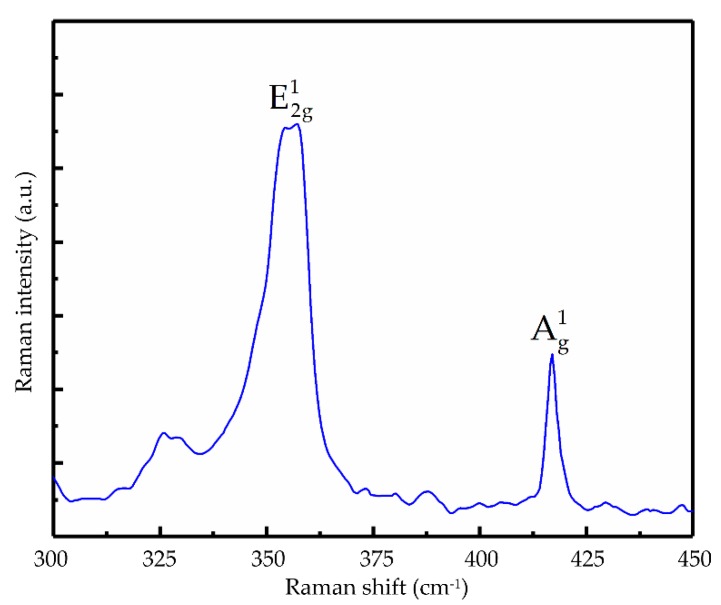
The Raman spectrum.

**Figure 4 nanomaterials-09-01200-f004:**
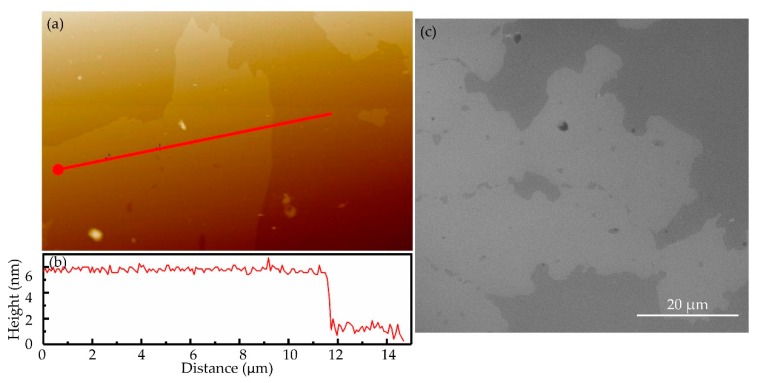
(**a**) Image of the atomic force microscope (AFM); (**b**) the thickness of the WS_2_ films; (**c**) image of the scanning electron microscope (SEM).

**Figure 5 nanomaterials-09-01200-f005:**
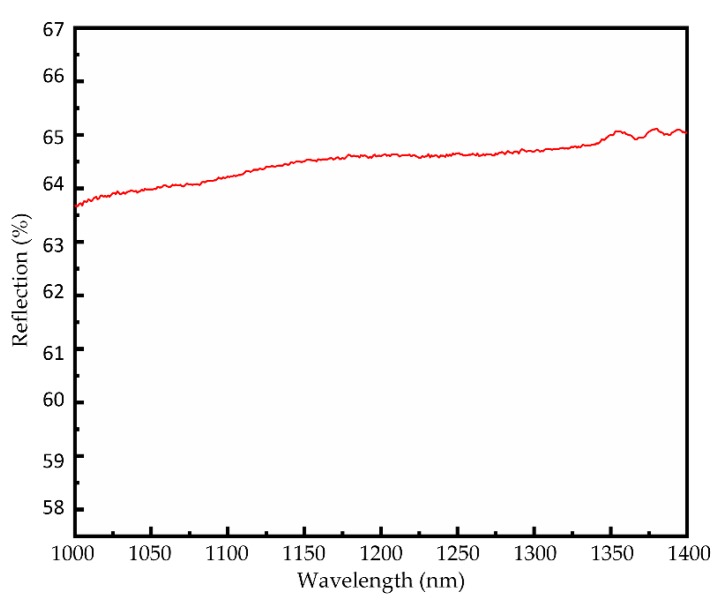
The reflectance spectrum of the reflective WS_2_ saturable absorber.

**Figure 6 nanomaterials-09-01200-f006:**
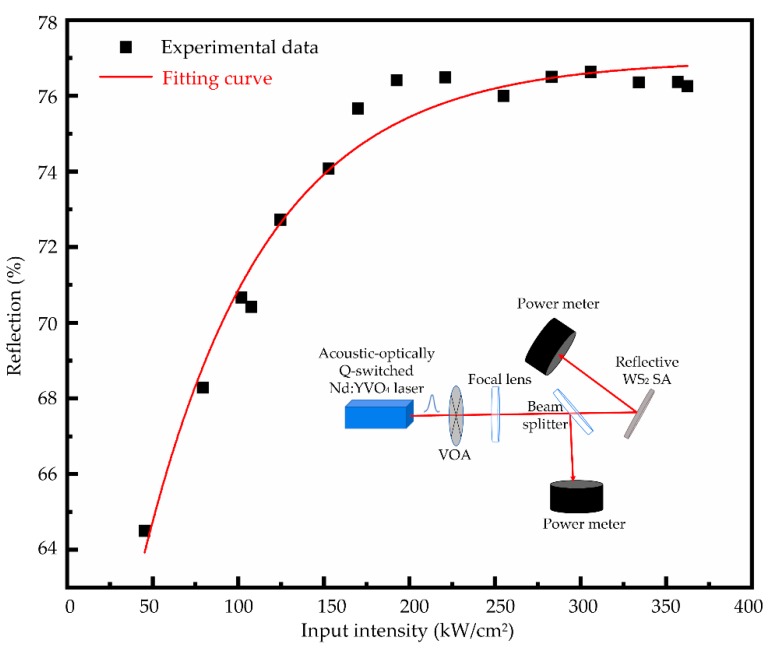
The nonlinear absorption saturation characteristics. Inset: the schematic diagram of nonlinear optical absorption measurement.

**Figure 7 nanomaterials-09-01200-f007:**
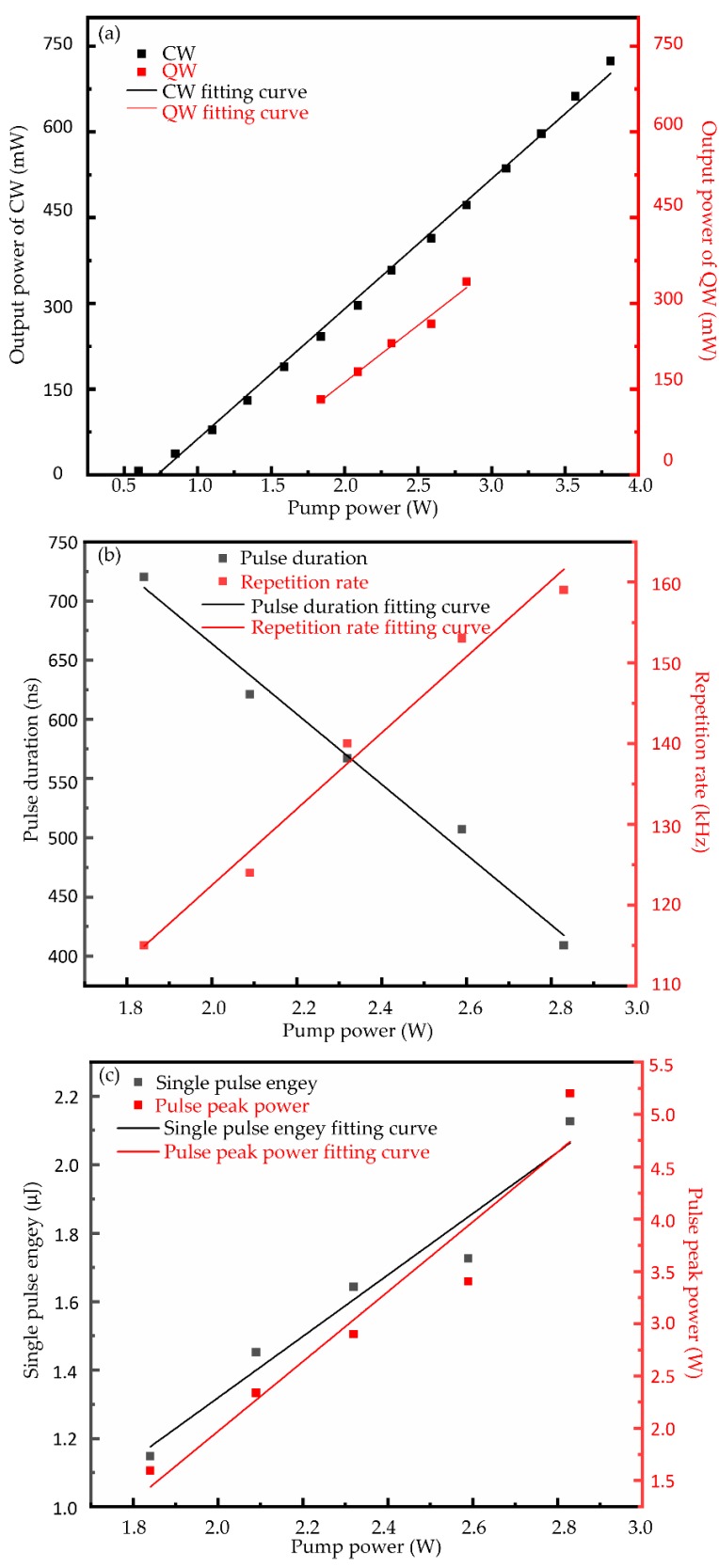
(**a**) The average output power of the continuous wave (CW) laser and of the Q-switched (QW) laser versus the pump power; (**b**) evolutions of the pulse duration and the pulse repetition rate with the pump power; (**c**) evolutions of the single pulse energy and the pulse peak power with the pump power.

**Figure 8 nanomaterials-09-01200-f008:**
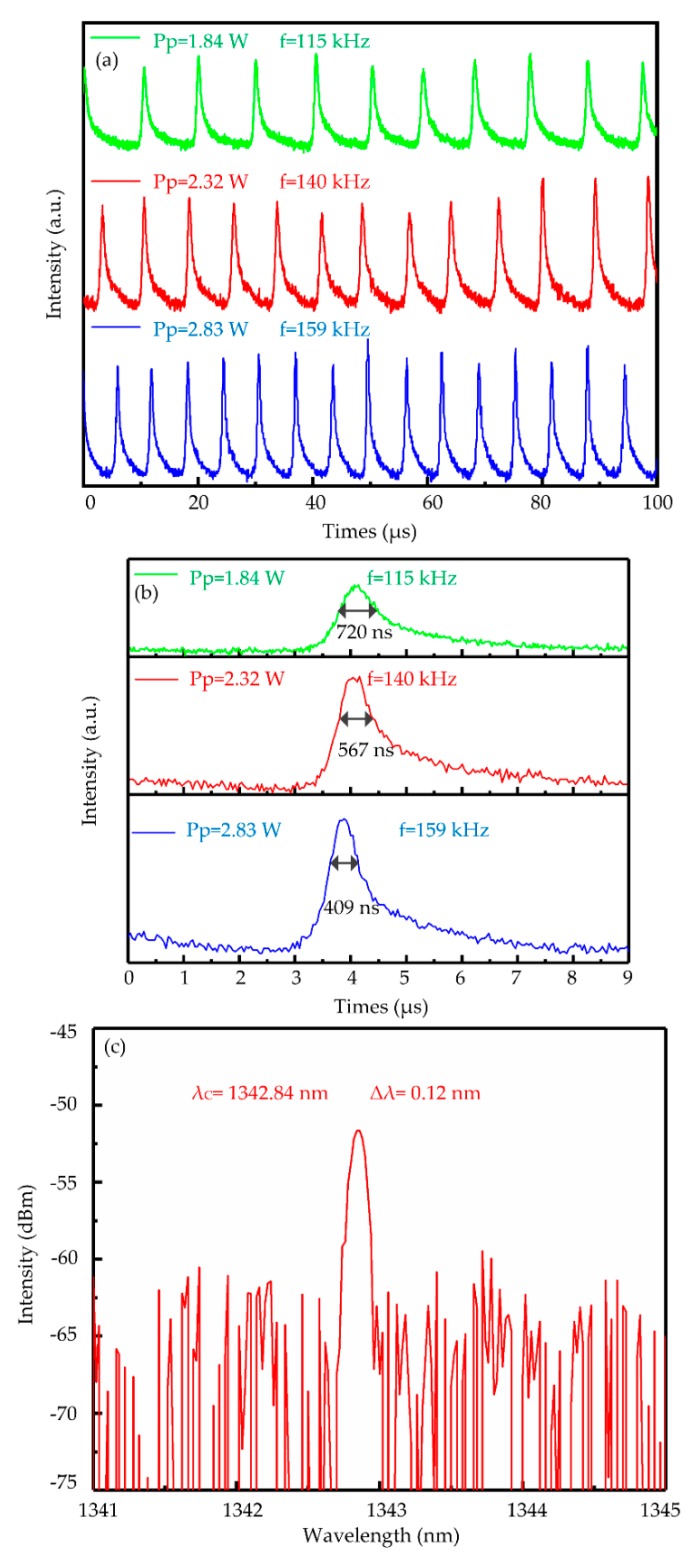
(**a**) The pulse trains of Q-switched lasers under different pump power; (**b**) the individual pulse under different pump power; (**c**) the Q-switched laser spectrum.

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
