# Peer review of "1.34 µm Q-Switched Nd:YVO4 Laser with a Reflective WS2 Saturable Absorber"

_nanomaterials, 2019, doi:10.3390/nano9091200_

Round 1

Reviewer 1 Report

The paper describes a passively Q-switched Nd:YVO4 laser with a few layers of WS2 created by Langmuir Blodgett film as a saturable absorber. I think the paper needs more introduction to WS2 as a material. I could not work out how the WS2 formed a layer, as it is not clear whether it is molecular or crystalline or ....? I also want to understand something about the chemical/ photo- stability of WS2 in the context of a laser cavity with high-intensity light.  

Again in the description of the WS2 layer fabrication, I don't know how large are the particles of WS2 that are exfoliated? You have done SEM so you should know that. Can you control the thickness of the deposited film? Can you deposit a second layer? or multiple layers? would that affect the behaviour of the laser? Have you modelled how this works to get a sense of what thickness and absorption is useful for a Q-switched laser at this power level? How do you know what cavity spot-size to use? Did you work it out?

I am frustrated by the list of measurement methods that you give, yet you don't show the results until later. But you also show the laser cavity in Fig 2, so I expect to see the other measurements. This figure ordering is illogical and I don't think it helps the reader.

Page 3/4 - what is the reference to a "frequency difference"? Is the figure showing a difference spectrum or a straight Raman spectrum of the WS2 layer on silver? what does the spectrum of silver look like? is there evidence of a SiO2 spectrum? In the discussion can you infer anything from the spectral width of the Raman peaks?

Page 4 Fig 4 needs improved contrast - really hard to discern what I am supposed to be seeing.

What is the average thickness of the WS2 films? what is the surface roughness? what is the surface variation, or thickness variation in the films?

page 5 is the spottiness of the right-hand side of fig 5 due to silver deposition or something else? You should comment on that.

page 5 what does the reflectance spectrum look like for just the silver mirror coating without WS2?

page 6 I don't think the Q-wave terminology is standard. I would refer to it as a Q-switched laser/

page 6 clarify if the pump power quoted in figures and text is incident on the mirror or on the crystal or absorbed or ....?

page 7 your figures should not join the data points with straight lines. You should use smooth curves or fit straight lines - whatever is physically reasonable. 

page 8 Fig 9c needs a vertical scale

page 9 how physically/chemically robust is the WS2 layer? does it rub off with your finger? does it ablate? does it need to be baked on?

page 9 what do you mean on line 155? do you have evidence for the thermal instability due to the atomic structure of WS2? is there a reference for this? why do you claim this?

also, do you have evidence that the thermal lens of the laser crystal causes instability?

Throughout the paper - too many acronyms - take out acronyms from figure captions and from the text where possible. They hinder understanding. (eg SA might be familiar to you but not to me.)

All the figures need better captions - they are too brief, use acronyms and don't tell me what I am seeing.

Page 1 check Ref line 29.

Page 2. line 46 what wafer?

Page 2 line 54 what is "instillation"?

Page 2 Silver mirror in figure is missing a letter.

Page 2 line 71 It is really confusing that you refer to the pump source as acousto-optically Q-switched when the whole paper is about saturable absorber Q-switch. The Acousto optically Q switched laser is used in a specific measurement, but that is not clear from the way you wrote the sentence.

Page 3 line 88 what is the "counterpart " side of the cavity? does not make sense.

Page 6 line 134 dont say "obviously" - you are insulting the reader.

page 9 line 179 check ref

page 10 line 228 check ref.

Reviewer 2 Report

Dear Authors,

from my point of view, the main purpose of this manuscript is not clear and it does not give any additional information to the international scientific comunity.

1)         WS2 saturable absorber is not a novelty in solid-state laser physics. It was used by Wenjing Tang two years ago. Please, see W. Tang et al. 2017,  IEEE Photonics Technology Letters 29(5), 2017.

2)         Nd:YVO4 is not a new gain material for laser applications. Please, see R. A. Fields et al., Appl. Phys. Lett. 51, 1885 (1987).

3)         The laser performances of the built-up Q-switched Nd:YVO4 are quite low if compared to those reported in literature. I skip listing all scientific papers focus on Nd:YVO4 as the Authors can find them on scholar-google or other scientific database.

Other comments:

1)         The listed bibliography does not reflect the large literature on the topic.

2)         If the scope of this manuscript is the fabrication of the WS2 by LB-method I think the experiment is not correctly presented.

For these reasons I believe that it does not match the criteria for publication in the Nanomaterials journal.

Reviewer 3 Report

I support publication of the paper. This is one more example of 2D semiconductor use in practical devices. 

But, I have one important concern. Namely, quality and presentation of one of their main results in Figure 9. They report that they use 1 ns risetime detector. 

What is the oscilloscope bandwidth and its input impedance? It seems to me that authors can not clearly resolve pulse train and individual pulses.

What is the reason that an individual pulses on Fig. 9a have such a long tail (equivalent to 3-5 microsec decay time). Pulse tail seems decaying faster  in Fig 9b.

Is pulse asymmetry scaling reciprocally with the amount of pump power above Q-switching mode threshold?

Could they provide individual pulse data at different pump levels (e.g. pulse trace at P=1.8W and 2.8W)? 

Other comments/concerns: How did they make sure that the intracavity lasing mode hits WS2 single/few layer flake? What happens when you shift the absorber sample to thicker areas? Any change in T vs incident power data (Fig. 7) when sample is shifted from one area to another? Also, 9 (c) figure needs a left scale labels (in dB). What is the full scale swing? or S/N ratio in Fig 9c data?

Round 2

Reviewer 1 Report

The authors have made a reasonable number of changes to the Manuscript, but they have ignored some of the suggestions. I am not sure if replotting graphs is too difficult, but I think they should relabel the QW in the graphs of Fig 7 and they should not join the dots in a piecewise continuous way. 

Reviewer 2 Report

Dear Authors,

The manuscript has been improved. I really appreciate your efforts.

However, I don't think it should be published. After reading it I didn't learn anything compared to what I already knew.

Again, the reference reported in the manuscript do not reflect the literature in any way. It is self-referential. Furthermore, it does not take into account "many and many" scientific efforts made by various international groups in the last 10 years.

Reviewer 3 Report

The manuscript has been improved along the lines suggested in the previous round.

Author Response

Great thanks for the reviewer’s comments and suggestions. Great thanks for the reviewer’s support for our scientific work and this manuscript. We expect that the revised paper can meet the requirement.